# Perspective of Immune Checkpoint Inhibitors in Thymic Carcinoma

**DOI:** 10.3390/cancers13051065

**Published:** 2021-03-03

**Authors:** Kyoichi Kaira, Hisao Imai, Hiroshi Kagamu

**Affiliations:** Department of Respiratory Medicine, Comprehensive Cancer Center, International Medical Center, Saitama Medical University, 1397-1 Yamane, Hidaka, Saitama 350-1298, Japan; m06701014@gunma-u.ac.jp (H.I.); kagamu19@saitama-med.ac.jp (H.K.)

**Keywords:** thymic carcinoma, immunotherapy, immune checkpoint inhibitor, PD-1 blockade, pembrolizumab, nivolumab

## Abstract

**Simple Summary:**

Thymic carcinoma is a rare neoplasm with a poor outcome, and there are no established therapeutic regimens for metastatic or recurrent disease. Immune checkpoint inhibitors (ICIs), such as PD-1/PD-L1 antibodies, are approved in several human cancers, however, ICIs are not approved in thymic carcinoma. Thus, several clinical trials have been undertaken to demonstrate if they are therapeutically effective for patients with thymic carcinoma. In our review, three prospective phase II studies and several case series were discussed in thymic carcinoma. We found that the objective response rate, disease control rate, and progression-free survival in PD-1 blockade monotherapy were approximately 20%, 73%, and four months, respectively. The therapeutic efficacy of PD-1 blockade monotherapy is still limited in patients with thymic carcinoma. Future perspectives focus on the therapeutic implication of tyrokinase inhibitors plus ICIs or new experimental agents plus ICIs alongside several ongoing experimental studies.

**Abstract:**

Thymic carcinoma is a rare neoplasm with a dismal prognosis, and there are no established therapeutic regimens for metastatic or recurrent disease. Immune checkpoint inhibitors (ICIs), such as PD-1/PD-L1 antibodies, are widely approved in several human cancers, contributing to prolonging survival in thoracic tumors. Thymic carcinoma exhibits histologic properties of squamous cell carcinoma (SQC), and resembles the SQC of the lung. ICIs are not approved in thymic carcinoma. Thus, several clinical trials have been undertaken to demonstrate if they are therapeutically effective for patients with thymic carcinoma. In our review, three prospective phase II studies and several case series were discussed in thymic carcinoma. We found that the objective response rate, disease control rate, and progression-free survival in PD-1 blockade monotherapy were approximately 20%, 73%, and four months, respectively. Two exploratory investigations indicated that PD-L1 within tumor cells exhibits a possibility of the therapeutic prediction of PD-1 blockade in thymic carcinoma. Several case reports, alongside their treatment content, have also been reviewed. The therapeutic efficacy of PD-1 blockade monotherapy is still limited in patients with thymic carcinoma. Future perspectives focus on the therapeutic implication of tyrokinase inhibitors plus ICIs or new experimental agents plus ICIs alongside several ongoing experimental studies.

## 1. Introduction

Thymic carcinoma (TC) is a rare neoplasm with a poor outcome. If early detected, surgical resection is suitable for its curability, whereas systemic chemotherapy (platinum-based regimens) is usually indicated for patients with metastatic or recurrent disease. However, chemotherapy yielded limited benefits to the outcome and efficacy of TC. Currently, there are no established regimens for the treatment of patients with advanced TC.

In fact, the assessment of chemotherapeutic agents for TC is limited to single-arm phase II trials or retrospective studies with small samples. Recently, Okuma et al. reported a systemic review and pooled analysis of anthracycline-, carboplatin-, or cisplatin-based chemotherapy in patients with TC [1]. Their results revealed that the objective response rates (ORRs) of anthracycline-based and non-anthracycline-based chemotherapy for advanced TC were 41.8% and 40.9%, respectively (*p* < 0.91), whereas those of cisplatin-based and carboplatin-based chemotherapy were 53.6% and 32.8%, respectively (*p* = 0.0029) in 206 patients collecting 10 studies. They concluded that cisplatin-based chemotherapy was better in patients with TC than carboplatin-based chemotherapy. Anthracycline-based regimens, such as ADOC (doxorubicin, cisplatin, vincristine, and cyclophosphamide), CODE (adriamycin, cisplatin, vincristine, and etoposide) and carboplatin plus amrubicin, and non-anthracycline-based chemotherapy, such as carboplatin plus paclitaxel, cisplatin plus docetaxel, and cisplatin plus irinotecan, are selected based on the judgment of chief-physicians. In fact, promising drugs, such as molecular targeting agents or immunotherapy aside from anthracycline or platinum-based regimens, are not available, although clinical trials have been conducted in different countries.

Recently, immune checkpoint inhibitors (ICIs), such as anti-programmed death-1 (PD-1)/programmed death ligand-1 (PD-L1) antibodies, have been administered to patients with several different kinds of neoplastic types. In particular, PD-1 blockade (nivolumab, pembrolizumab, and atezolizumab) show significant efficacy in patients with advanced non-small-cell lung cancer (NSCLC). Varying efficacies of these antibodies according to PD-L1 expression within tumor cells have been reported. PD-1 blockade monotherapy, or a combination of PD-1 blockade with platinum-based regimens, have been identified as first-line standard treatments [2,3,4,5]. By recent investigation, Kaplan-Meier curves of overall survival (OS) exhibited an estimated 5-year rate of 34.2% among patients with melanoma, 27.7% among patients with renal cell carcinoma, and 15.6% among patients with NSCLC [6]. Although the possibility of long-term survivors after PD-1 blockade treatment has been confirmed in several reports [7,8], ICIs have been identified as a curative modality in limited cancer types.

Unfortunately, the results of several trials could not strongly approve the clinical utilization of PD-1 blockade in patients with metastatic or recurrent TC. Therefore, we reviewed the clinical evidence of ICIs and discussed the perspective of this strategy in such patients.

## 2. Materials and Methods

### 2.1. Selection Criteria for Literature

We researched the papers published in English using the MEDLINE and PubMed databases to identify all reviews, clinical studies, and case reports about the relationship between thymic carcinoma and immune checkpoint inhibitors. The search strategy included articles published between March 2018 and November 2020 using the following keywords: “thymic carcinoma”, “immunotherapy”, “immune checkpoint inhibitor”, “PD-1”, “nivolumab”, “pembrolizumab”, “atezolizumab”, “durvarumab”, and “avelumab”. We also researched the abstracts to identify reports of low activity that would not appear in the published literature. The search did not restrict the type of publication or periodical. We did not include preliminary results published as abstracts or meeting proceedings. We selected all published reports describing the clinical significance of immune checkpoint inhibitors in TC.

### 2.2. Study Selection

We found 12,044 papers from searching “thymic carcinoma”, 372 papers from searching the combination of “thymic carcinoma” and “immunotherapy”, 46 papers from searching the combination of “thymic carcinoma” and “immune checkpoint inhibitor”, 46 papers from searching the combination of “thymic carcinoma” and “PD-1”, 23 papers from searching the combination of “thymic carcinoma” and “pembrolizumab”, 11 papers from searching the combination of “thymic carcinoma” and “nivolumab”, 0 papers from searching the combination of “thymic carcinoma” and “durvarumab”, and 3 papers from searching “thymic carcinoma” and “avelumab”. Criteria for exclusion were insufficient data on constructed 2 × 2 contingency tables and duplicate studies on the same patients. The final decision about inclusion was based on the full article.

## 3. Results

This section may be divided by subheadings. It should provide a concise and precise description of the experimental results, their interpretation, as well as the experimental conclusions that can be drawn.

### 3.1. Demographics of Selected Literature

Among 372 papers, 15 papers were selected as novel reports for our review. There were three review articles, four original articles, six case reports, and two others.

### 3.2. Prospective Study

Three prospective clinical trials focused on 10 patients with thymic carcinoma receiving PD-1 blockade monotherapy [9,10,11]. The therapeutic demographics of these clinical trials are listed in Table 1.

Pembrolizumab: Cho et al. conducted a phase II study to evaluate the overall response rate (ORR) of pembrolizumab as the primary endpoint in 33 patients with refractory or relapsed thymic epithelial tumors [9]. Among the 33 patients, 26 had TC, and 7 had thymoma. The ORR and disease control rate (DCR) of the thymoma cases were 28.6% and 100%, respectively, while the ORR, DCR, median progression-free survival (PFS), and median overall survival (OS) in TC were 19.2%, 73.1%, 6.1 months, and 14.5 months, respectively. In TC patients, 12 (46.2%) had already received radiotherapy for the primary mediastinal mass, 9 (34.6%) underwent curative-intent surgery for the tumor, and 11 (42.3%) were treated with pembrolizumab as a first-line treatment, after previous chemotherapy. All enrolled patients were Korean. Immune-related adverse events (irAEs) in the TC patients included grade 1–2 thyroiditis, pruritus, skin rash, grade 3–4 hepatitis, myasthenia gravis, and subacute myoclonus in 1 (3.8%), 3 (11.5%), 2 (7.7%), 2 (7.7%), 2 (7.7%), and 1 (3.8%), respectively. In addition, they examined the relationship between the efficacy of pembrolizumab and PD-1 immunostaining in 24 patients (72.7%); however, little is known about the number of patients with TC histology. High PD-L1 expression (≥50% tumor proportion score) was observed in 14 (58.3%) patients. Among the 24 patients with high PD-L1 expression, 5 (35.7%) achieved partial response (PR), whereas none of the patients with low PD-L1 expression yielded a response (*p* = 0.034). There was no association between the frequency of irAEs and the status of PD-L1 expression. This study found a significant correlation between PD-L1 protein expression and mRNA expression. The results of this study indicated that pembrolizumab monotherapy is active and tolerable with manageable irAEs for patients with advanced TC, and the expression level of PD-L1 may predict the efficacy of pembrolizumab using an exploratory approach.

Giaccone et al. also reported a phase 2 study to evaluate the efficacy of pembrolizumab in patients with TC [10]. In their study, 40 patients were eligible, and the median follow-up period was 20 months. Nineteen (48%) had a histology of squamous cell carcinoma, four (10%) were Asian, and 33 (82%) were Caucasian. Twenty-three (58%) patients received previous chest radiotherapy, and 21 (52%) were treated with a previous thymectomy. The ORR and DCR yielded 22.5% and 75.0%, respectively. One patient achieved a complete response (CR), 8 (20%) PR, 21 (53%) stable disease (SD), and 10 progressive disease (PD), without any cases of pseudoprogression. The median PFS, median OS, 1-year PFS, and 1-year OS were 4.2 months, 24.9 months, 29% and 71%, respectively. The most common grade 3 or 4 adverse events were increased aspartate aminotransferase and alanine aminotransferase (13% patients, each). Moreover, six (15%) patients developed one or more severe irAEs. In particular, 2 (5%) patients experienced polymyositis and myocarditis, requiring high-dose steroid therapy. In this study, PD-L1 immunostaining was investigated in 37 patients. High PD-L1 expression (≥50% tumor proportion score) was observed in 10 (25%) patients, 6 of whom achieved a PR or CR (5 PR and 1 CR) (*p* = 0.005). Among the 27 patients with low PD-L1 expression, 23 had PD. Their post-hoc analysis demonstrated that median PFS was longer in patients with high PD-L1 expression than in those with low expression (24 months vs. 2.9 months), and OS in patients with high expression was longer than in those with low expression (not reached vs. 15.5 months). In subsequent analysis, 5 of 40 eligible patients completed two years of treatment, and four of them elected to continue treatment, with more than four years [11].

Nivolumab: Katsuya et al. reported a phase II study of nivolumab in 15 patients with unresectable or recurrent TC to evaluate ORR as the primary endpoint [12]. Thirteen patients had a histology of squamous cell carcinoma. Seven (46.6%) patients received prior radiotherapy, and prior surgery was performed in three (20%) patients. The median follow-up period was 14.1 months. The ORR, DCR, median PFS, and median OS were 0%, 73.3%, 3.8 months, and 14.1 months, respectively. Eleven patients achieved SD, and four had PD. The subgroup analysis of 13 patients with squamous cell carcinoma indicated that the median PFS and OS were 3.8 months and 14.1 months, respectively. Moreover, the median PFS and OS in seven patients with previous radiation therapy were 3.8 months and 12.7 months, respectively. The profile of adverse events was mild consistent with previous data. Among the two patients who experienced serious irAEs, grade 3 transaminase increase was observed in one, and grade 2 adrenal insufficiency was observed in the other requiring admission.

### 3.3. Exploratory Study

Ak et al. explored their experience with nivolumab for metastatic thymic epithelial tumors [13]. Among the 46 unresectable and recurrent thymic epithelial tumors (32 thymoma and 14 TC), 8 (3 TC, 4 thymoma, and one mixed histology of thymoma and TC) received nivolumab. Table 2 shows the clinical features of four patients with TC. Among them, two achieved an ORR of PR, but the median PFS of these two patients was 2.0 and 5.8 months, respectively. There was PD-L1 expression with more than 50%. Among them, one patient experienced severe irAEs, such as dyslexia and motor neuropathy, requiring the withdrawal of nivolumab.

### 3.4. Case Reports

We found five case reports indicating the unique clinical course of patients with TC who received PD-1 blockade [14,15,16,17,18]. The summary of these case reports is displayed in Table 3.

Jin et al. reported two cases of refractory thymic squamous cell carcinoma receiving nivolumab or pembrolizumab combined with chemotherapy/angiogenetic therapy [14]. Case 1 received nivolumab as a second-line therapy, sequentially, re-administration of nivolumab combined with nab-paclitaxel as a fifth-line therapy. The OS from initiation of nivolumab was more than three years. Case 2 was treated with pembrolizumab as a second-line therapy, with PD. Sequentially, combination therapy with anlotinib plus pembrolizumab was continued for 16 cycles with SD. After PD, the patient experienced PR, switching to combination with pembrolizumab plus gemcitabine. The duration of pembrolizumab and its combination therapy was 23 months. Cafaro et al. described a case of heavily pretreated TC who was treated with pembrolizumab as the fourth-line treatment [15]. Uchida et al. demonstrated an experience of nivolumab in four cases of unresectable TC [16]. Case 1 was treated with carboplatin plus nab-paclitaxel as a first-line treatment, and radiotherapy was initiated as palliative therapy. The patient then received nivolumab as a second-line therapy, and PR was found after five cycles. Case 2 also received nivolumab after first-line treatment with carboplatin plus nab-paclitaxel. After four cycles of nivolumab, there was evidence of PR. In case 3, carboplatin plus paclitaxel, as a first-line treatment, was continued until four cycles, then sequentially with gemcitabine for 44 cycles. Nivolumab was administered as a third-line treatment, and radiological assessment showed PR after four cycles. Case 4 received five cycles of carboplatin plus paclitaxel as a first-line therapy and six cycles of gemcitabine as a second-line therapy. Nivolumab was chosen as a third-line treatment, and six cycles were completed with a response of SD. Yang et al. described a better response to nivolumab as a fourth-line therapy in a patient with TC [17]. Isshiki et al. documented a successful case of refractory thymic carcinoma with high PD-L1 expression treated with pembrolizumab as a second-line therapy after three cycles of carboplatin plus nab-paclitaxel [18].

### 3.5. Clinicopathological Relevance of PD-L1 Expression in Thymic Carcinoma

The expression of PD-L1 within tumor cells is associated with the therapeutic prediction of PD-1 blockade in human cancers, especially NSCLC or head and neck cancer [2,19,20]. Although several descriptions regarding the relationship between prognostic significance and PD-L1 expression in patients with different types of cancers have already been reported, several researchers also documented whether the expression of PD-L1 could be related to tumor aggressiveness and survival in patients with TC [21,22,23,24,25,26]. Recently, Katsuya et al. reported the prognostic relevance of PD-L1 expression in patients with 102 thymomas and 37 thymic carcinomas [21]. The positive rate of PD-L1 expression was identified as 70% in TC and 23% in thymoma, with a significant difference (*p* < 0.001). However, there was no significant difference in OS between positive and negative PD-L1 expression in TC. In another report, high expression of PD-L1 was observed in 25 (36%) patients with TC, though no significant difference in the PFS and OS was observed between patients with high and low PD-L1 expression [22]. Weissferdt et al. also did not find any prognostic impact of high PD-L1 expression, although expression of PD-L1 was recognized in 14 (54%) patients with TC [23]. Moreover, a clinicopathological study of 20 patients with TC revealed a trend of better OS in patients with low PD-1 expression, without a significant difference [24]. On the other hand, Yokoyama et al. demonstrated the opposite results of the prognostic relevance of PD-L1 expression compared to those of several studies above [25]. In their study, PD-L1 was highly expressed in 20 (80%) patients with TC, and low PD-L1 expression was identified as a significant predictor of worse survival [24]. However, Funaki et al. reported that PD-L1 was highly expressed in 17 (39.5%) patients with TC, and positive PD-L1 expression was closely associated with worse PFS and OS after surgical resection [26]. To date, some studies have analyzed the prognostic relevance of PD-L1 expression in all patients with TC and thymoma [27,28]. The expression level of PD-L1 displayed a higher trend in TC than in thymoma; however, they appeared to be conflicting results regarding whether PD-L1 could predict better or worse outcomes. Thymoma complicates autoimmune disorders (myasthenia gravis) and includes immature T-cells expressing terminal deoxynucleotidyl transferase in tumor tissues, whereas TC is known to lack immature T-lymphocytes and does not induce autoimmune diseases [29,30]. As the tumor immune environment is different between thymoma and TC and TC also exhibited significantly worse survival than thymoma, both diseases should be distinguished in the survival analysis. From previous studies, we found that PD-L1 is highly expressed in TC, but it remains unclear whether the expression level of PD-L1 predicts the outcome after any treatment. The difference in the clone of PD-L1 antibody for immunohistochemistry and study design may have biased the results of each investigation. Although a small sample size is dominant owing to the rarity of the disease, further study is required to evaluate the prognostic implications of PD-L1 expression in TC using a large-scale approach. The reagents and scoring criteria used for the assessment of PD-L1 above studies are listed in Table 4.

## 4. Discussion

Currently, PD-1 blockade is approved in patients with several types of human neoplasms. In particular, it has been found to improve the therapeutic efficacy of PD-1 blockade combined with cytotoxic agents or angiogenetic drugs [4]. Therefore, we expect that PD-1 blockade is therapeutically effective for patients with TC as rare cancer. In our review, three prospective phase II studies and several case series regarding TC were discussed, and we found that the ORR, DCR, and median PFS in PD-1 blockade monotherapy were approximately 20%, 73%, and four months, respectively. Compared with the clinical trial of nivolumab in previously treated NSCLC, the therapeutic outcome of PD-1 blockade monotherapy in TC seemed favorable, and the increased frequency of irAEs was not found. Although the expression of PD-L1 within tumor cells is known to predict the efficacy of PD-1 blockade in patients with NSCLC, two exploratory investigations indicated that PD-L1 has the potential to precisely predict PD-1 blockade in TC [9,10]. Cho et al. and Giaccone et al. described that the clinical benefit according to the expression level of PD-L1 seems to be apparent for predicting the efficacy of PD-1 blockade in patients with thymic carcinoma [9,10]. Several case reports have also been reviewed, and the treatment content discussed. In clinical practice, nivolumab or pembrolizumab is active for patients with previously treated TC. Moreover, their clinical effectiveness was confirmed to persist in the long-term. Further studies are required to confirm the efficacy of PD-1 blockade according to the expression level of PD-L1.

To improve the therapeutic efficacy of TC, the development of new regimens, such as tyrosine kinase inhibitors (TKIs), ICIs, and their combination, is necessary, and many studies are ongoing. As clinical trials of PD-1 blockade monotherapy in patients with thymic epithelial tumors, three phase II studies are currently ongoing. Atezolizumab was investigated in 34 patients with pretreated TC, with the primary endpoint of ORR and secondary endpoint of PFS, OS, duration of objective response (DOR), DCR, adverse events, and distribution of PD-L1 and TMB expression (NCT04321330). Nivolumab was examined in 55 patients with previously treated TC and thymoma type B3 after platinum-based chemotherapy, with the primary endpoint of PFS rate at six months and secondary endpoint of PFS, OS, ORR, DCR, and DOR (NCT03134118). Avelumab was investigated for 55 patients with previously treated TC and thymoma after platinum-based chemotherapy, with a primary endpoint of ORR and safety and secondary endpoint of immune-related PFS, OS, and DOR (NCT03076554). The PFS, OS, and DOR were common in these three studies, but primary outcome measures were different among these studies, PFS or ORR.

In addition, combination therapy of TKIs with PD-1 blockade has also been examined as a targeted treatment for thymic epithelial tumors. A phase II study evaluating the ORR of pembrolizumab plus sunitinib is currently ongoing in 40 patients with TC who are resistant to platinum-based regimens (NCT03463460). The primary objective of this study is ORR and secondary objectives are to evaluate safety profile, PFS and OS. As exploratory investigations, it is planning to determine whether sunitinib leads to an increase in PD-L1 expression, TILs, and decrease in myeloid-derived suppressor cells (MDSC) in both tumor and peripheral blood. In the CAVEATT study, 33 patients with TC and thymoma type B3 treated with platinum-based chemotherapy are registered and received avelumab combined with axitinib (an oral VEGFR-1/2/3 kinase inhibitor) for the primary endpoint of ORR. In phase I/II trials, nivolumab plus vorolanib, an oral VEGFR/PDGFR kinase inhibitor, is in progress for the aim of safety and ORR in 177 patients with pretreated thoracic tumors, including TC (NCT03583086). The primary objectives of this study are safety and tolerability of nivolumab plus vorolanib as a phase I study and efficacy as a phase II study, and secondary objectives are PFS and OS. Finally, pembrolizumab with or without SC-C101 (a superagonist fusion protein of interleukin-15) is currently ongoing as a phase 1/Ib study for 96 patients with pretreated solid tumors, including thymic epithelial tumors (NCT04234113). The primary objective of this study is to determine dose-limiting toxicities and treatment-related adverse events, and secondary objectives are ORR, BOR, PFS, and clinical benefit rate. These clinical trials are promising and are expected to provide future perspectives.

## 5. Conclusions

Very few prospective studies have evaluated the efficacy of PD-1 blockade monotherapy (pembrolizumab or nivolumab) in patients with pretreated TC. The therapeutic efficacy of PD-1 blockade monotherapy is still limited in such patients, similar to the therapeutic results of advanced NSCLC. Although it remains unclear whether PD-L1 expression could predict the efficacy of PD-1 blockade monotherapy in TC, an exploratory investigation suggests an increased response of PD-1 blockade in patients with high PD-L expression. We believe that PD-L1 appears predictive in such patients. Future perspectives focusing on the therapeutic implication of TKIs plus ICIs or new experimental agents plus ICIs, and several experimental studies are currently ongoing.

## Figures and Tables

**Table 1 cancers-13-01065-t001:** Demographics of prospective phase 2 study of PD-1 blockade in thymic carcinoma.

Different Variables	Cho et al. [9]	Giaccone et al. [10]	Katsuya et al. [11]
Patient’s number	26	40	15
Median age			
years (range)	57 years (26–78)	57 years (25–80)	55 years (34–70)
Gender			
Male/Female	18/8 (69.2%/30.8%)	28/12 (70%/30%)	12/3 (80%/20%)
Race			
Asian	26 (100%)	4 (10%)	15 (100%)
Caucasian	0 (0%)	33 (82%)	0 (0%)
others	0 (0%)	3 (8%)	0 (0%)
ECOG PS			
0–1/2	22 (100%)/0 (0%)	38 (96%)/2 (5%)	15 (100%)/0 (0%)
Prior use of chemotherapy	26 (100%)	40 (100%)	15 (100%)
ORR (%)	19.2%	22.5%	0.0%
DCR (%)	73.1%	75.0%	73.3%
PFS (months)	6.1 months	4.2 months	3.8 months
OS (months)	14.5 months	24.9 months	14.1 months
1-year PFS rate (%)	NA	29.0%	9.0%
1-year OS rate (%)	NA	71.0%	60.0%
Previous radiotherapy	12 (46.2%)	23 (58%)	7 (73.3%)
Curative surgery	9 (34.6%)	21 (52%)	3 (20%)
irAEs Grade 1–2 Grade 3–4	Thyroiditis (3.8%) Pruritus (11.5%) Skin rash (7.7%)	Fever (13.8%) Hypothyroidism (13%) Rash (10%)	Rash (27%) Hypothyroidism (7%) Diarrhea (20%)
	Hepatitis (7.7%) Myasthenia gravis (7.7%) Subacute myoclonus (3.8%)	ALT/AST increased (13%) Myocarditis (5%) Myalgia or Myositis (8%)	AST increased (7%)

Abbreviation: PD-1, programmed death-1; ECOG, eastern clinical oncology group; PS, performance status; ORR, objective response rate; PFS, progression-free survival; OS, overall survival; irAEs, immune-related adverse events; ALT, alanine aminotransferase; AST, aspartate aminotransferase; NA, not applicable.

**Table 2 cancers-13-01065-t002:** Clinical features of four patients with thymic carcinoma (exploratory study).

Patient	Gender	Age	Histology	PD-L1 (%)	Treatments before ICI	Nivolumab Treatment				Survival before ICI (Months)	Survival after ICI (Months)	Total OS (Months)
						Total N of Cycles	Best Response	PFS (Months)	irAEs			
No. 1	M	60	SCC	60	Pac/Carbo, Etoposide, CAP	3	SD	3.2	none	41.2	22.1	63.3
No. 2	M	26	SCC, T	100	Pac/Carbo	3	PR	2.0	Myasthenic Symptoms Grade 1	70.2	6.5	76.7
No. 3	F	67	SCC	100	CAP	1	NA	NA	Dislexia, motor neuropathy Grade 4	7.5	6.6	14.1
No. 4	F	73	SCC	50	Pac/Carbo	10	PR	5.8	none	10.5	6.2	16.7

Abbreviations: PD-L1, programmed death ligand-1; ICI, immune checkpoint inhibitor; N, number; PFS, progression-free survival; OS, overall survival; SCC, squamous cell carcinoma; T, thymoma; Pac, paclitaxel; Carbo, carboplatin; CAP, cisplatin/adriamisin/cyclophosphamide; SD, stable disease; PR, partial response; NA, not applicable. Nivolumab was administered intravenously at 3 mg/kg or 240 mg/day every two weeks.

**Table 3 cancers-13-01065-t003:** Summary of case series regarding PD-1 blockade treatment in thymic carcinoma.

Case Series	Age	Sex	Treatment Line	PD-1 Blockade or Combination Regimens (Response)	Complete Cycles or Treatment Duration	irAEs
Jin et al. [14]						
Case 1	40	M	2nd 5th	Nivolumab (SD) Nivolumab plus nab-paclitaxel (PR)	40 cycles ^†^ 38 months ^‡^ 3 cycles	G1 skin
Case 2	52	M	2nd 3rd 4th	Pembrolizumab (PD) Pembrolizumab plus anlotinib (SD) Pembrolizumab plus gemcitabine (PR)	2 cycles ^§^ 16 cycles 23 months ^‡^	Gr3 skin
Cafaro et al. [15]						
Case 1	57	M	4th	Pembrolizumab (SD)	18 cycles	NA
Uchida et al. [16]						
Case 1	72	F	2nd	Nivolumab (PR)	NA	NA
Case 2	64	F	2nd	Nivolumab (PR)	NA	NA
Case 3	66	M	3rd	Nivolumab (PR)	8 cycles	NA
Case 4	72	F	3rd	Nivolumab (PR)	6 cycles	NA
Yang et al. [17]						
Case 1	62	F	4th	Nivolumab (PR)	NA	G1 skin
Isshiki et al. [18]						
Case 1	68	F	2nd	Pembrolizumab (PR)	NA	none

Abbreviation: PD-1, programmed death-1; irAEs, immune-related adverse events; G, grade; SD, stable disease; PR, partial response; PD, progressive disease; NA, not available; M, male; F, female; 1st, 1st line treatment; 2nd, 2nd line treatment, 3rd, 3rd line treatment; 4th, 4th line treatment. ^†^ Nivolumab and ^§^ pembrolizumab were administered intravenously at 3 mg/kg or 240 mg/day every two weeks and 200 mg/day every three weeks, respectively. ^‡^ This time is defined as time to progressive disease.

**Table 4 cancers-13-01065-t004:** Reagents and scoring criteria used for the assessment of PD-L1.

Reference	Clone of PD-L1 AntiBody	Company, City, Country	Definition of Scoring Criteria
[9]	22C3	Dako North America, Carpinteria, CA, USA	PD-L1 expression was classified as “high” if at least 50% of the tumor cells, inflammatory cells, or stroma cells stained positive. PD-L1 expression in 0% to 49% of cells was classified as “low” expression.
[13]	28-8	Abcam, Tokyo, Japan	PD-L1 ≥ 1% was defined as “positive”.
[10]	22C3	Agilent Technologies, Santa Clara, CA, USA	PD-L1 expression was classified as “high” if at least 50% of the tumor cells stained positive. PD-L1 expression in 1–49% of cells was classified as “low” expression.

Abbreviation: PD-L1, programmed death ligand-1.

## Data Availability

No new data were created or analyzed in this study. Data sharing is not applicable to this article.

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
