# Peer review of "Perspective of Immune Checkpoint Inhibitors in Thymic Carcinoma"

_cancers, 2021, doi:10.3390/cancers13051065_

Round 1

Reviewer 1 Report

The authors provide a review on the status of PD-L1 and immune therapies in Thymic carcinoma. They present data from several small and large studies to document the frequency of expression in these tumors. In addition, they review the published data on the use of I/O in this tumors. 

Comments

1) The authors should provide a table of the reagents and scoring criteria used for the assessment of PDL1. As the authors are no doubt aware these can introduce significant differences in the results.

2) The authors focus a bit too much on the case reports; this part of the presentation could be shortened.

3) The authors should provide greater details of the current/ ongoing trials and highlight the similarities and differences in these. Currently this important information is covered in a single paragraph. 

Author Response

Reviewer 1

The authors provide a review on the status of PD-L1 and immune therapies in Thymic carcinoma. They present data from several small and large studies to document the frequency of expression in these tumors. In addition, they review the published data on the use of I/O in this tumors. 

Comments

  • The authors should provide a table of the reagents and scoring criteria used for the assessment of PDL1. As the authors are no doubt aware these can introduce significant differences in the results.

Re) According to reviewer’s suggestion, a table of the reagents and scoring criteria used for the assessment of PD-L1 was provided as Table 3. Therefore, previous Table 3 was changed to Table 4.

  • The authors focus a bit too much on the case reports; this part of the presentation could be shortened.

Re) According to reviewer’s suggestion, the descriptions of case series were shortened.

3) The authors should provide greater details of the current/ ongoing trials and highlight the similarities and differences in these. Currently this important information is covered in a single paragraph. 

Re) According to reviewer’s suggestion, we provided more detailed information regarding the current/ongoing trials.

Reviewer 2 Report

Perspective of Immune Checkpoint Inhibitors in Thymic Carcinoma

Abstract : rate (line 20)

In all the document where applicable: Median progression-free survival instead of PFS.

Lines 21-26: English to improve.

Introduction:

English to review: examples: line 34, 48-51, 54, 57-59, 65.

Methods:

For completeness, I would have searched the abstracts as well – would potentially identify reports of low activity that would not appear in the published literature.

English to review: examples: line 77, 87-89

Results:

English to review: examples: lines 116-117, 124-125, 145-146, 161, 165, 187-188, 193-194, 198-199, 209, 240-241

No need to repeat the exact information found in the table (refer to table 1 instead).

Line 122: please confirm that you are talking about PD-L1 mRNA expression.

Lines 140-141: how many PR/CRs ??

Line 150: 7 divided by 15 is 46,6%.

Line 157-159: side effect profile seems very low. Do you check with the authors to validate this information?

Line 166: PD-L1 > 50% in all 4 patients?

Line 181: TTP instead of PFS

Line 221: define positive (> 1%?, > 50%?)

Lines 214-240: This section is confusing. Please specify which antibody was used for PD-L1 testing. What cut-off for a positive score was used? Maybe better to present this section as another table.

Lines 252-273: this section is irrelevant to this clinical review. To be removed.

Table 1:

review English. Provide information on prior use of chemotherapy.

Considering that median PFS is a very poor marker of efficacy with ICIs, perhaps looking at 1- and 2- year progression free survival rate would allow us to better appreciate the tail of the curve effect.

Table 2:

In individual patients, should refer to time to progressive disease instead of PFS.

Number of cycles: please specify – 4 week cycles ? Doses every 2 weeks?

Table 3:

Not only demographics are represented. Please change the title.

This table is not very useful – should include duration of response or time to PD for all patients.

There is definitely a publication bias in favor of responsive patients…

Discussion:

English to review: lines 276-278, 283-287, 294, 301-302, 312-313, 320

The demonstration has not been done in this paper to support effectiveness in SD (line 289)

Author Response

Reviewer 2

Abstract : rate (line 20)

In all the document where applicable: Median progression-free survival instead of PFS.

Lines 21-26: English to improve.

 Re) According to reviewer’s suggestions, our manuscript was corrected.

Introduction:

English to review: examples: line 34, 48-51, 54, 57-59, 65.

 Re) According to reviewer’s suggestions, our manuscript was corrected.

Methods:

For completeness, I would have searched the abstracts as well – would potentially identify reports of low activity that would not appear in the published literature.

English to review: examples: line 77, 87-89

 Re) According to reviewer’s suggestions, our manuscript was corrected. The following sentence was added in methods; “We also researched the abstracts as well to identify reports of low activity that would not appear in the published literature.”

Results:

English to review: examples: lines 116-117, 124-125, 145-146, 161, 165, 187-188, 193-194, 198-199, 209, 240-241

Re) According to reviewer’s suggestions, our manuscript was corrected.

No need to repeat the exact information found in the table (refer to table 1 instead).

Re) Thank you for your generous comments.

Line 122: please confirm that you are talking about PD-L1 mRNA expression.

Re) This sentence indicates the relationship between PD-L1 protein and mRNA.

Lines 140-141: how many PR/CRs ??

Re) CR was observed in one patient and PR in 5 patients. This information was added.

Line 150: 7 divided by 15 is 46,6%.

Re) Thank you for your generous comments. We made a mistake, so, 46.6% was corrected.

Line 157-159: side effect profile seems very low. Do you check with the authors to validate this information?

Re) We checked the profile of adverse events again. Because of small numbers, authors may report the frequency of side effects.

Line 166: PD-L1 > 50% in all 4 patients?

Re) As described by Table 2, all patients exhibited PD-L1 more than 50%.

Line 181: TTP instead of PFS

Re) According to reviewer’s suggestions, our sentence was corrected.

Line 221: define positive (> 1%?, > 50%?)

Re) Definition of PD-L1 positive finding was listed in Table 4.

Lines 214-240: This section is confusing. Please specify which antibody was used for PD-L1 testing. What cut-off for a positive score was used? Maybe better to present this section as another table.

Re) We are sorry that we didn’t present the outline of PD-L1 testing. Table 4 was added in this section.

Lines 252-273: this section is irrelevant to this clinical review. To be removed.

Re) According to reviewer’s suggestions, this section including reference 31, 32 was removed from this manuscript.

Table 1:

review English. Provide information on prior use of chemotherapy.

Considering that median PFS is a very poor marker of efficacy with ICIs, perhaps looking at 1- and 2- year progression free survival rate would allow us to better appreciate the tail of the curve effect.

 Re) According to reviewer’s suggestions, above information available was added in Table 1.

Table 2:

In individual patients, should refer to time to progressive disease instead of PFS.

Number of cycles: please specify – 4 week cycles ? Doses every 2 weeks?

Re) According to reviewer’s suggestions, the administration of nivolumab was added in Table 2. However, there was no information regarding time to progressive disease.

Table 3:

Not only demographics are represented. Please change the title.

This table is not very useful – should include duration of response or time to PD for all patients.

There is definitely a publication bias in favor of responsive patients…

Re) Title was changed, but, the information of duration of response or time to PD for all patients was not available.

Discussion:

English to review: lines 276-278, 283-287, 294, 301-302, 312-313, 320

The demonstration has not been done in this paper to support effectiveness in SD (line 289)

Re) According to reviewer’s suggestions, “even if SD was achieved” was deleted.

Round 2

Reviewer 2 Report

I wish to thank the authors for their review and improvements.

Nonetheless, some information is still difficult to understand from my reading of this current version:

1 ) evaluation of benefit according to PD-L1 seems apparent from the evaluation by Cho and Giaccone but the specific paragraph discussing the importance the importance of PD-L1 as a predicitive/prognostic marker does not discuss those two studies (the data is lost elsewhere in the text). I would propose that all the data relevant to PD-L1 expression be united in the relevant section in order to have a better grasp of this aspect. The conclusion is that PD-L1 appears predictive but this is not apparent from reading the PD-L1 section alone as it currently reads.

2) The section on current trials should be written in the present tense as those trials are ongoing and have not been reported so far. Will be much easier to read.

Author Response

Thank you for your generous comments.

I revised our paper points by points using red font.

I wish to thank the authors for their review and improvements.

Nonetheless, some information is still difficult to understand from my reading of this current version:

1 ) evaluation of benefit according to PD-L1 seems apparent from the evaluation by Cho and Giaccone but the specific paragraph discussing the importance the importance of PD-L1 as a predicitive/prognostic marker does not discuss those two studies (the data is lost elsewhere in the text). I would propose that all the data relevant to PD-L1 expression be united in the relevant section in order to have a better grasp of this aspect. The conclusion is that PD-L1 appears predictive but this is not apparent from reading the PD-L1 section alone as it currently reads.

Re) According to reviewer’s suggestions, we added the comment and discussion of predictive potential of PD-L1 expression in the discussion (Line, 133-139, 159-162, 282-285, 332).

2) The section on current trials should be written in the present tense as those trials are ongoing and have not been reported so far. Will be much easier to read

Re) Thank you for your generous comments. We corrected the past tense to the present tense.
